# The Role of Sodium-Glucose Co-Transporter-2 Inhibitors on Diuretic Resistance in Heart Failure

**DOI:** 10.3390/ijms25063122

**Published:** 2024-03-08

**Authors:** Panagiotis Stachteas, Athina Nasoufidou, Dimitrios Patoulias, Paschalis Karakasis, Efstratios Karagiannidis, Michail-Angelos Mourtzos, Athanasios Samaras, Xanthi Apostolidou, Nikolaos Fragakis

**Affiliations:** 1Second Department of Cardiology, Aristotle University of Thessaloniki, Hippokration General Hospital of Thessaloniki, Konstantinoupoleos 49, 54642 Thessaloniki, Greece; panstacha@auth.gr (P.S.); athinaso@auth.gr (A.N.); pakar15@hotmail.com (P.K.); stratoskarag@gmail.com (E.K.);; 2Outpatient Department of Cardiometabolic Medicine, Second Department of Cardiology, Aristotle University of Thessaloniki, 54642 Thessaloniki, Greece; patoulias@auth.gr

**Keywords:** SGLT-2 inhibitors, acute decompensated heart failure, diuretics, fluid balance, congestion

## Abstract

Heart failure (HF) remains a major cause of morbidity and mortality worldwide. Recently, significant advances have been made in its treatment; however, diuretics remain the cornerstone in managing congestion in HF. Although diuretic resistance poses a significant challenge in the management of HF and is associated with poor outcomes, only limited alternative pharmaceutical options are available in clinical practice. The objective of this narrative review is to provide a comprehensive analysis of the current evidence on the effects of sodium-glucose co-transporter-2 (SGLT-2) inhibitors on diuretic resistance in HF patients. The primary emphasis is placed on clinical data that assess the impact of SGLT-2 inhibitors on fluid balance, symptom improvement, and clinical outcomes and secondarily on safety profile and potential adverse effects associated with SGLT-2 inhibitor use in acute decompensated HF. The current evidence on the efficacy of SGLT-2 on diuretic resistance remains controversial. Findings from observational and randomized studies are quite heterogenous; however, they converge on the notion that although SGLT-2 inhibitors show promise for mitigating diuretic resistance in HF, their diuretic effect may not be potent enough to be widely used to relieve objective signs of congestion in patients with HF. Importantly, the introduction of SGLT-2 inhibitors in HF treatment appears to be generally well tolerated, with manageable adverse effects. Further research is needed to investigate the underlying mechanisms and the possible beneficial impact of SGLT-2 inhibitors on diuretic resistance in HF.

## 1. Introduction

Heart failure (HF) is a chronic disorder of the cardiovascular (CV) system and remains a major cause of morbidity and mortality worldwide [1,2]. Recently, significant advances have been made in its treatment [1,2]. In recent decades, although the prevalence of HF with reduced ejection fraction (HFrEF) has stabilized in the majority of developed Western countries, the prevalence of HF with mildly reduced (HFmrEF) and preserved ejection fraction (HFpEF) in both developed and developing countries is constantly increasing [3]. Therefore, the cumulative prevalence of the disease is constantly increasing at an alarming rate, reaching the level of 3–20 patients per 1000 people in the general population [2]. This is due to the greater number of patients surviving after acute myocardial infarction (AMI), but also due to the increase in life expectancy and the increasing prevalence of traditional CV risk factors (e.g., arterial hypertension, obesity, dyslipidemia, and type 2 diabetes mellitus (T2DM)) in relatively younger people [4]. It is important to note that HF remains the leading cause of premature death in patients with established cardiovascular disease (CVD) worldwide, irrespective of the prevailing clinical phenotype [5]. Furthermore, in addition to negatively impacting patients’ quality of life and functional capacity [6], HF places a significant economic burden on health systems as it is associated with high healthcare costs, partly due to frequent hospitalizations and readmissions [2].

Individuals suffering from acute or chronic HF often encounter breathlessness, whether at rest or during physical activity. This is typically accompanied by elevated cardiac filling pressures, increased circulated levels of natriuretic peptides, and evidence of fluid overload such as swelling, ascites, lung crackles, or pleural effusion [2]. Many of these symptoms and clinical signs are linked to the kidneys retaining sodium and water [7]. Consequently, the use of diuretics plays a pivotal role in managing fluid balance in HF providing the alleviation of symptoms and relief from physical signs of congestion [8,9].

The inability to attain effective congestion relief with low urinary sodium levels, even when employing sufficient or increasing diuretic dosages, is termed as diuretic resistance [7,10]. Diuretic resistance (DR) is recognized as a firmly established contributor to the deterioration of heart failure, prolonged hospitalization, elevated rates of readmission, and heightened morbidity and mortality [7,10]. The standard approach to address this challenging issue in HF management involves introducing a distinct diuretic agent like thiazides [11], acetazolamide [12], spironolactone [13], or tolvaptan [14] that can inhibit sodium reabsorption in a different segment of the nephron simultaneously [15].

The latest evidence from landmark trials suggests that sodium-glucose co-transporter-2 (SGLT-2) inhibitors are beneficial for patients suffering from T2DM or established CVD, mainly HF, by reducing CVD-related morbidity and mortality [16,17]. Notwithstanding the significant amount of evidence in the field, the precise underlying mechanisms that underpin these advantageous outcomes are still not fully understood. Thus, the actual impact of SGLT-2 inhibitors on diuretic resistance in HF patients remains controversial.

The aim of this narrative review is to provide a comprehensive analysis of the current evidence on the effects of SGLT-2 inhibitors on diuretic resistance in HF patients. The focus is primarily on clinical data from observational and randomized studies that assess the impact of SGLT-2 inhibitors on decongestion in patients with fluid overload and symptom improvement and secondarily on the safety profile and potential adverse effects associated with SGLT-2 inhibitor use in decompensated HF.

## 2. Diuretic Resistance in HF

HF exacerbation is mainly associated with symptoms from excess extracellular fluid volume. The development of acute decompensated HF leads to poor outcomes and an impaired quality of life. Even now, diuretics are the cornerstone of acute decompensated HF treatment and the most commonly described symptomatic drug therapy for chronic congestive HF when decongestion is necessary [18,19]. Patients with HF are frequently hospitalized and more than 50% exit the hospital without sufficient weight loss and with residual congestion. Although there is not a universally accepted definition, the failure to achieve appropriate congestion relief with low urine sodium concentration, despite using adequate or escalating diuretic doses, is described as diuretic resistance (DR) [7,10]. DR is considered a well-established factor of worsening HF, the prolongation of hospital stays, higher readmission rates, and increased morbidity and mortality [7,10]. 

In patients with acute decompensated or chronic congestive HF, DR derives from a variety of mechanisms [20] (Figure 1). One well-established mechanism is the compensatory post-diuretic sodium reabsorption (CPDSR) and post-diuretic retention [18]. High doses of loop diuretics induce the remodeling of the distal nephron, including hypertrophy and hyperplasia of the distal convoluted tubule, the connecting tubule, and the collecting ducts. When administering loop diuretics for long periods, sodium absorption is obstructed in the loop of Henle, but may be enhanced in the hypertrophied distal tubule, resulting in an unchanged sodium balance [18]. Moreover, the initial fluid loss is followed by the activation of the renin–angiotensin–aldosterone system (RAAS) and of the sympathetic nervous system (SNS), resulting in post-diuretic sodium retention and mitigating the beneficial outcomes of loop diuretics [18,20]. 

Another possible pathophysiologic mechanism of DR in HF is the diuretic braking phenomenon [21]. This phenomenon arises from the long-term administration of a specific dose of diuretic medication, leading to a gradual decline in peak natriuresis over time. This decline can be partially attributed to structural changes (remodeling) in the nephrons induced by the prolonged use of high doses of loop diuretics. Specifically, the dose–response curve in congestive HF shifts downwards and to the right (Figure 1), indicating a reduction in the maximum achievable natriuresis and an increase in the dose of diuretics required to achieve similar diuretic effects [21]. As in post-diuretic retention, the braking phenomenon is inextricably intertwined with neurohormonal activation and RAAS activation. Additionally, it is well-established that loop diuretics can initiate renin secretion through the obstruction of the sodium–potassium–chloride cotransporter, by enhancing prostacyclin production and by inducing volume contraction [22]. Overall, the diuretic braking phenomenon represents a complex interplay of structural and biochemical changes within the kidney, neurohormonal activation, and RAAS activity, contributing to the development of DR in patients with HF.

Pre-tubular mechanisms are also involved in the pathogenesis of DR in HF [23]. A negative sodium balance is mandatory for HF patients. A low sodium intake of 80–120 mmol is considered ideal, because only in this instance, sodium excretion exceeds dietary intake [23]. Chronic kidney disease (CKD) and reduced renal reperfusion caused by low cardiac output, hypotension, or central venous congestion impairs the glomerular filtration rate (GFR), further deteriorating the secretion of the loop diuretics into the tubule lumen. Furthermore, nonsteroidal anti-inflammatory drugs (NSAIDs), uremic anions, uric acid, and metabolic acidosis suspend diuretic secretion, by competing for the same ion transporters [23]. Decreased blood flow in the intestine or mucosal edema can limit the absorption of loop diuretics; however, hypoproteinemia seems to be significant only in severe conditions such as nephrotic syndrome [23]. In addition, hypochloremia is considered an alternative pathophysiological mechanism involved in DR in HF, as chlorine was identified as the main ion related to DR because of its complicity in renal sodium sensing, delivering pathways, and in renin release according to another study [24]. Finally, in recent studies, gene polymorphisms of the sodium-chloride cotransporter (NCC), of the subunits of epithelial sodium channel transporters (EnaC), or of the organic anion transporters (OATP1B1) seem to be involved in diuretic responsiveness and resistance [7].

Addressing diuretic resistance can be extremely challenging in HF management. The primary consequence is the inadequate control of fluid retention. Increasing diuretic doses can complicate treatment by potentially inducing electrolyte imbalances and exacerbating renal failure, which, in turn, might further deteriorate the underlying medical conditions. Diuretic resistance necessitates more attentive management, including frequent dose adjustment, the combination of diuretics and other drugs, and regular blood testing, while looking out for adverse outcomes, complications, and the worsening of HF in situ. An enhanced understanding of diuretic resistance and identifying individual patient-specific factors can contribute to effectively addressing this significant challenge. 

## 3. SGLT-2 Inhibitors, Mechanism of Action, and Current Clinical Indications

SGLT-2 inhibitors represent a recently developed group of oral medications for diabetes management. They function by impeding the reabsorption of both sodium and glucose in the kidneys, achieved through reducing the renal threshold for glucose absorption within the proximal tubule. This action results in the excretion of glucose and sodium in the urine, leading to glycosuria (glucose in urine) and natriuresis (sodium in urine) [25]. SGLT-2 inhibitors lower plasma glucose levels in individuals with diabetes, but their effect on plasma glucose levels is negligible in those with normal glycemic levels [26]. Apart from their glucose-lowering effects, they have been associated with body weight loss, blood pressure (BP) reduction, and a lower risk of hypoglycemia compared to other commonly prescribed antidiabetic drugs, like insulin and sulfonylureas [17,26,27].

In clinical settings, SGLT-2 inhibitors are employed for the management of T2DM as a second-line therapeutic option subsequent to insufficient glycemic regulation with metformin, or as a primary treatment for individuals at high risk for CVD, HF, or CKD [28,29,30,31]. Furthermore, they have recently obtained authorization for use in patients with HF across the entire spectrum of left ventricular ejection fraction (LVEF) independent of glycemic control, as per the most recent guidelines outlined by the American Heart Association [32] and the 2023 ESC guidelines for the management of cardiovascular disease in patients with diabetes [33].

This indication stems from cumulative evidence demonstrating their advantageous impact on decreasing HF-related hospitalizations and cardiovascular mortality rates [34], thereby establishing a novel foundational role for SGLT-2 inhibitors in HF management. Beyond their well-documented cardiovascular benefits, SGLT-2 inhibitors exhibit a renoprotective profile independent of kidney disease severity or etiology, as well as diabetes status [35]. However, despite the pleiotropic benefits [36], their impact on the mitigation of diuretic resistance in patients with acute decompensated HF remains less clear.

## 4. The Intersection of SGLT-2 Inhibitors and Diuretic Resistance

### 4.1. Pathophysiology and Mechanisms

The diuretic effect of SGLT-2 inhibitors occurs in the proximal convoluted tubule (PCT), where SGLT2s are located. SGLT-2 inhibitors reduce sodium reabsorption by causing the constriction of the proximal arteriole and the dilatation of the distal arteriole, lowering the GFR. SGLT-2 inhibitors also reduce proteinuria and hyperfiltration [20]. Because the majority of sodium is reabsorbed in the loop of Henle and the distal tubule, SGLT-2 inhibitors have minimal diuretic effects, but they can enhance the diuretic response when combined with other classes of diuretics by improving the responsiveness to atrial natriuretic peptide [20].

Preliminary clinical data from trials using SGLT-2 inhibitors in patients with acute HF needing diuresis are encouraging. In a double-blind, randomized, controlled trial, empagliflozin increased urinary output only for four days in ADHF patients, but it reduced mortality, HF rehospitalization, and in-hospital-worsening HF at 60 days [37]. In a small retrospective study recruiting 31 patients with T2DM who received SGLT-2 inhibitors as adjuvant therapy, weight loss, urine volume, and diuretic efficiency was improved 24 h after initiation, without the worsening of renal function, potassium, or blood pressure [38]. The diuretic synergy of dapagliflozin and bumetanide was evaluated in healthy subjects, in whom the diuretic effect and Na^+^ excretion was enhanced when one drug treatment was added on the other after a week of single-drug treatment [39]. 

Τhe underlying mechanisms through which SGLT-2 inhibitors may mitigate diuretic resistance are multifactorial and involve intricate physiological processes (Figure 2). These pharmacological agents exert their effects through a cascade of interconnected pathways, each contributing to the overall therapeutic outcome. Their effects on natriuresis, renal function, SNS activation, and hemodynamics play a pivotal role [40]. Firstly, SGLT-2 inhibitors exhibit a pronounced impact on natriuresis, the excretion of sodium in the urine, thereby promoting the elimination of excess fluid from the body. By inhibiting the reabsorption of glucose and sodium in the proximal renal tubules, these agents enhance the urinary excretion of both substances, leading to a net reduction in extracellular fluid volume. Moreover, the renal effects of SGLT-2 inhibitors extend beyond simple natriuresis. They modulate renal function by influencing the GFR and tubuloglomerular feedback mechanisms. By enhancing the GFR and altering tubular dynamics, these agents contribute to the overall regulation of fluid balance and renal hemodynamics. Animal studies have also presented evidence of the suppression of the SNS by moderating the adrenergic activity of the afferent sympathetic nerve, resulting in the reduced activation of the RAAS [40]. SGLT-2 inhibitors have an additional distinctive attribute to shrink the interstitial fluid volume greater than the intravascular fluid volume. This mechanism acts protectively against the neurohormonal activation induced by alterations in intravascular fluid volume [40].

Furthermore, SGLT-2 inhibitors have shown undeniably beneficial effects on the kidney. In an experimental study involving rats, dapagliflozin exhibited a remarkable reduction in inflammatory processes and fibrotic changes within the nephron, indicating a potential mechanism mediated by the activation of sirtuin-1 (SIRT1) and subsequent inhibition of nuclear factor-κB (NF-κB) expression. This intricate pathway not only mitigates immediate renal damage but also holds promise for attenuating the progression of chronic kidney disease by curbing long-term oxidative stress [41].

Moreover, SGLT-2 inhibition offers additional advantages through its effects on erythropoiesis and hematocrit levels. It has been proposed that SGLT-2 inhibitors modulate medullary oxygen tension, which in turn influences the function of myofibroblasts within the renal interstitium. This alteration in myofibroblast function promotes the synthesis and release of erythropoietin, leading to an elevation in erythropoietin levels and subsequent increase in hematocrit. The rise in hematocrit not only enhances oxygen-carrying capacity systemically, but also plays a crucial role in improving oxygen delivery specifically to the renal cortex and medulla. This enhancement in renal tissue oxygenation is vital for maintaining optimal renal function and may contribute to the preservation of renal structure and function over the long term [41].

### 4.2. Efficacy Data

Several clinical studies have investigated the impact of SGLT-2 inhibitors on the attenuation of diuretic resistance in patients with HF (Table 1). SGLT-2 inhibitors seem to have a moderate diuretic effect (increases in urine output), a result that has been observed repeatedly when they are integrated in the conventional treatment of HF patients with loop diuretics [42,43,44]. This diuretic effect is mainly attributed to the combination of induced glycosuria causing osmotic diuresis and the resultant activation of compensatory mechanisms, which in turn halts the diuretic effect of SGLT-2 inhibitors [39,44] and occurs mostly at the early stages of treatment. This effect diminishes around the 24 h mark after the initiation of SGLT-2 inhibitors and is not sustained in the long term [38,39,42,43,44,45,46,47]. These findings are promising for the use of SGLT-2 inhibitors as an effective add-on therapy to standard diuretic medication among patients hospitalized with ADHF.

Moreover, the diuresis itself seems to not be linked with natriuresis, as most short-term studies show no significant alterations in urinary sodium excretion and the evidence does not support a sustained natriuretic effect of SGLT-2 inhibitors [38,39,42,43,44,45,46,47]. There is only one study of 20 euvolemic HF patients with DR, which documented a more sustained natriuretic effect of empagliflozin that remained for up to 2 weeks, which led to a significant decrease in blood volume [39]. Interestingly, some studies report that when empagliflozin was used in combination with bumetanide, there was a compound effect in respect to fractional sodium excretion in urine, suggesting a synergistic effect of these drugs to natriuresis [39]. These results are in agreement with the mechanism of action of SGLT-2 inhibitors, which presents them more as modulators of volume that help the nephron manage sodium and fluid more efficiently, rather than as pure diuretic agents [48].

Apart from their impact on fluid balance, SGLT-2 inhibitors lead to significant decreases in body weight (BW) when included in conventional therapy, compared to a placebo [39,42,49]. However, it is not yet clear that the observable decreases in BW are completely due to decongestion [44]. Some trials suggest that the decline in BW could be a result of a loss of calories in urine, because of the glycosuria caused by SGLT-2 inhibitors [38,44]. For instance, according to findings from the EMPA-RESPONSE-AHF trial, despite the elevation in urinary output, empagliflozin was not associated with decongestion signs (either with improved symptoms of dyspnea (*p* = 0.18) or improved diuretic response (−0.35 ± 0.44 vs. −0.12 ± 1.52 kg/40 mg furosemide equivalents; *p* = 0.37)) over the first 4 days compared to the control group [38]. 

Concomitantly, with their effects on fluid balance, SGLT-2 inhibitors elicit elevations in hematocrit levels among HF patients when incorporated into conventional therapeutic regimens, as evidenced by multiple trials [44,46,48]. These investigations indicate that the mechanism underlying this hematocrit increase predominantly stems from heightened erythropoietin production and subsequent erythropoiesis, rather than solely from hemoconcentration induced by diuresis and volume depletion [44,46,50,51].

In addition, SGLT-2 inhibitors improve the symptomatology of HF patients, when assessed using the KCCQ-TSS or other congestion scores [52,53]. These results were documented mostly 12 weeks after the beginning of treatment [44,53,54]. Interestingly, the results stemming from the EMPULSE trial suggest that the improvement in symptomatology in patients across the KCCQ-TSS can be documented as early as 15 or 30 days after the initiation of SGLT-2 inhibitors [52]. However, none of the aforementioned studies reported improvement in respect to objective markers of congestion such as jugular venous distension, edema, ascites, or pleural effusion [44], while those that did showed no significant changes [44,54,55]. The lack of objective or patient-reported markers of decongestion was further highlighted by the fact that no stable correlation was observed when examining the levels of natriuretic peptides (in particular NT-proBNP) alongside the improvement in clinical briefing when administrating an SGLT-2 inhibitor. Some studies reported no change regarding NT-proBNP levels [43,44,50,54,55], and on the other hand, a few studies reported a reduction in NT-proBNP [43,44,50,52]. The heterogeneity of the results in the given trials indicates that there is no solid evidence to support the theory that the beneficial effects of these drugs in the symptomatology of HF patients, or even in the reduction in the risk of death or hospitalization, are solely due to a distinct action in renal function and successful decongestion. This suggests that there may be a different underlying mechanism of action [44,56].

Other important findings referred to the dosage of loop diuretics, which generally required less intensification with the co-administration of SGLT-2 inhibitors compared to the control group [50,52,57]; however, at the same time, their impact on the frequency of the de-escalation of the baseline dosage or of the loop diuretic discontinuation between treatment arms was ambiguous [39,49,58]. Moreover, in one study, the possibility of adding another loop diuretic dropped significantly with the addition of an SGLT-2 inhibitor in the treatment of HF patients, and those results, along with other clinical benefits and the observed decreases in diuretics dosage, were independent of the type or the dosage of the diuretics used in the trial [58]. One possible underlying mechanism is the synergistic diuretic effect of SGLT-2 inhibitors when added on loop diuretics. Indeed, according to findings from a single-center, double-blind, placebo-controlled RCT (EMPAG-HF), the efficacy of furosemide was significantly increased in the SGLT-2 inhibitor arm (14.1 mL urine/mg furosemide equivalent, 95% CI: [0.6–27.7]; *p* = 0.041) [43]. However, when compared to other diuretics (e.g., metolazone), SGLT-2 inhibitors were less effective (non-significance) at relieving congestion when added to intravenous loop diuretics in patients with HF and DR (MD −0.08 kg, 95% CI: [−0.17–0.01]; *p* = 0.10) [47].

**Table 1 ijms-25-03122-t001:** Key studies investigating the impact of SGLT-2 inhibitors on the attenuation of diuretic resistance in patients with HF.

Study ID	Type of Study	Population (Main Characteristics)	SGLT2 Inhibitor vs. Comparator	Follow-Up	Main Outcomes
Yeoh et al., 2023 [47]	Multicenter, open-label, randomized, parallel group trial	61 patients hospitalized for HF with resistance to treatment with iv loop diuretics (furosemide)	Dapagliflozin 10 mg/day vs. metolazone 5–10 mg once daily. Randomization 1:1. Duration of treatment 3 days.	5 days	✓ The mean reduction in body weight (primary outcome) was 3.0 (±2.5) kg in the SGLT-2 inhibitor arm compared to 3.6 (±2.0) kg in metolazone group (MD = 0.65 kg, 95% CI: [−0.12–1.41]; *p* = 0.11). Loop diuretics were less efficient when administered with dapagliflozin than with metolazone (MD −0.08 kg, 95% CI: [−0.17–0.01]; *p* = 0.10). ✓ Changes in pulmonary congestion were similar in both groups. ✓ Dapagliflozin resulted in smaller reductions in plasma, sodium, and potassium levels and more moderate increases in urea and creatinine when compared to metolazone. ✓ Serious adverse events were similar in both groups.
Biegus et al., 2023 [52]	Prespecified secondary analysis of the multicenter, double-blind RCT (EMPULSE trial)	530 patients hospitalized due to symptoms and signs of ADHF requiring iv loop diuretics after initial stabilization	Empagliflozin 10 mg/day vs. placebo as add-on therapy for 3 months. Randomization 1:1.	90 days	✓ Empagliflozin was associated with significantly greater reductions in body weight compared to the control group at day 15 (adjusted MD = −1.97 kg, 95% CI: [−2.86 to −1.08], *p* < 0.0001), at day 30 (MD = −1.74 kg, 95% CI: [−2.73 to −0.74], *p* = 0.0007), and after 3 months (MD = −1.53 kg, 95% CI: [−2.75 to −0.31], *p* = 0.0137). ✓ Empagliflozin was not associated with the administration of greater doses of loop diuretics compared to the control group at day 15 (adjusted MD = 6.7 mg of iv furosemide (or equivalent), 95% CI: [−1.0 to 14.4], *p* = 0.0862), at day 30 (MD = 5.3 mg of iv furosemide (or equivalent), 95% CI: [−1.6 to 12.3], *p* = 0.1295), and after 3 months. (MD = 3.1 mg of iv furosemide (or equivalent), 95% CI: [ −3.4 to 9.6], *p* = 0.3488) ✓ Treatment with empagliflozin was associated with greater congestion score reductions compared to controls at day 15 (adjusted MD = −0.34, 95% CI: [−0.60 to −0.09], *p* < 0.01) and after 3 months (MD = −0.23, 95% CI: [−0.47 to 0.02], *p* = 0.067).
Chatur et al., 2023 [58]	Prespecified subgroup analysis of the DELIVER multicenter RCT	6263 patients with HFpEF and at least intermittent diuretic requirement, divided in three groups: (i) No-diuretic (10.9%), (ii) Non-loop diuretic (12.3%), (iii) Loop-diuretic (76.8%) (furosemide < 40 mg, 40 mg and >40 mg)	Dapagliflozin 10 mg/day vs. placebo. Randomization 1:1	3 years	✓ Dapagliflozin decreased the addition of a loop diuretic by 32% (HR 0.68, 95% CI: [0.55–0.84], *p* < 0.001), without reducing discontinuations (HR 0.98, 95% CI: [0.86–1.13], *p* = 0.83) in the follow-up. ✓ SGLT2 inhibitors compared to a placebo were associated with less sustained dose increases of furosemide (between group differences in the mean increase in the dose of furosemide −2.5 mg/year, 95% CI: [−1.5 to −3.7], *p* < 0.001) and a more pronounced decrease in the sustained dose of furosemide (95% CI: [−9.4 to −3.6%]; *p* < 0.001). ✓ There was no statistically significant difference in the treatment benefits or in the observed serious adverse effects of dapagliflozin across all subgroups of diuretic use (p-interaction = 0.64) and furosemide dosage (p-interaction = 0.57).
Charaya et al., 2022 [49]	Single-center, open-label, randomized pilot study	102 patients hospitalized for ADHF and requiring iv administration of loop diuretics, with LVEF between 30.2% and 59.6% and eGFR 32.1–71.1 mL/min	Dapagliflozin 10 mg/day in addition to standard diuretics vs. conventional therapy. Randomization 1:1	6 days intrahospital and 30 days after discharge	✓ Patients receiving dapagliflozin required, on average, lower doses of loop diuretics compared to the control group (mean dose 78.46 mg/day vs. 102.82 mg/day; *p* = 0.001) and displayed a decreased frequency of the up-titration of loop diuretics compared to the control group (14% vs. 30%; *p* = 0.048). However, the need to add another class of diuretics (thiazides or acetazolamide) did not differ (10% vs. 15%; *p* = 0.66). ✓ In the group receiving dapagliflozin, there was a decrease in the eGFR (−4.2 mL/min vs. 0.3 mL/min) after 48 h (primary outcome). However, no statistically significant difference in the eGFR was observed between the two groups on discharge (*p* = 0.36). ✓ The mean decrease in body weight was greater in the SGLT2 inhibitor arm (4.1 kg vs. 3 kg; *p* = 0.02). ✓ The rehospitalization rate as well as the mortality during hospitalization and 30 days after discharge were similar in both groups.
Mordi et al., 2020 [42]	Single-center, double-blind, placebo-controlled, crossover RCT (RECEDE-CHF trial)	23 patients with T2DM and HFrEF and furosemide dose of 18.3–80.9 mg/day	Empagliflozin 25 mg/day vs. placebo as add-on therapy to regular loop diuretic for 6 weeks. Then, 2 weeks as a washout period, and then drug switch between the two groups and treatment for 6 more weeks. Randomization 1:1.	6 + 6 weeks	✓ Patients receiving empagliflozin experienced a substantial increase in 24 h urinary volume at day 3 (MD = 535 mL, 95% CI: [133–936], *p* = 0.005) and week 6 (MD = 545 mL, 95% CI: [136–954], *p* = 0.005) (primary outcome); however, their 24 h urinary sodium excretion after 6 weeks did not significantly increase (MD = −7.85 mmol/L, 95% CI: [−2.43–6.73], *p* = 0.57). ✓ Patients in the SGLT2 inhibitor arm experienced a small non-significant elevation in the fractional excretion of Na after 3 days (MD = 0.30%, 95% CI: [−0.03–0.63]; *p* = 0.09), which attenuated later after 6 weeks of treatment (MD = 0.11%, 95% CI: [−0.22–0.44]; *p* < 0.99). ✓ After 6 weeks of empagliflozin treatment, a substantial increase in the clearance of electrolyte-free water (MD = 312 mL, 95% CI: [26–598]; *p* = 0.026) and a statistically significant decrease in body weight were observed (*p* < 0.001).
Schulze et al., 2022 [43]	Single-center, double-blind, placebo-controlled, RCT (EMPAG-HF)	60 patients hospitalized for ADHF and requiring loop diuretics administration	Empagliflozin 25 mg/day vs. placebo as add-on therapy to regular loop diuretic. Randomization 1:1.	5 days (efficacy) and 30 days (safety outcomes)	✓ Patients receiving empagliflozin experienced a 25% increase in cumulative urine excretion compared to the control group (median 10.8 vs. 8.7 L; group difference estimation 2.2 L, 95% CI: [8.4–3.6]; *p* = 0.003). ✓ The efficacy of furosemide was significantly increased in the SGLT2 inhibitor arm (14.1 mL urine/mg furosemide equivalent, 95% CI: [0.6–27.7]; *p* = 0.041). ✓ Markers of renal injury (levels of total urinary protein; mean urinary α1-microglobulin) and other safety outcomes were similar between the two groups. ✓ The levels of NT-proBNP showed significantly greater decrease in the group treated with empagliflozin (−1861 vs. −727.2 pg/mL; *p* < 0.001).
Damman et al., 2020 [39]	Multicenter, double-blind, placebo-controlled pilot study (EMPA-RESPONSE-AHF)	79 patients hospitalized for acute HF requiring loop diuretics administration	Empagliflozin 10 mg/day vs. placebo for 1 month. Randomization 1:1.	30 days	✓ Empagliflozin was associated with significantly increased urinary output up to day 4 compared to the control group (MD = 3449 mL, 95% CI: [578–6321]; *p* < 0.01). ✓ Empagliflozin was not associated with either improved symptoms of dyspnea (*p* = 0.18) or improved diuretic response (−0.35 ± 0.44 vs. −0.12 ± 1.52 kg/40 mg furosemide equivalents; *p* = 0.37) over the first 4 days compared to the control group. ✓ The length of the hospital stay was similar in both groups (*p* = 0.58). ✓ Empagliflozin was safe and well tolerated, and the adverse effects were similar in both groups.
Tamaki et al., 2021 [47]	Single-center, open-label RCT	59 patients with T2DM hospitalized for AHF requiring loop diuretics administration	Empagliflozin 10 mg/day vs. standard antidiabetic treatment. Randomization 1:1.	1 week	✓ Patients in the empagliflozin group experienced a greater total urine excretion (*p* = 0.005), as well as a greater urinary excretion of glucose (*p* < 0.001) and sodium (*p* = 0.015) during the first day. ✓ As far as other indices of decongestion, there was a significant decrease in the levels of NT-proBNP in the SGLT2 inhibitor arm (*p* = 0.04), a significant decrease in plasma volume (*p* = 0.017), and a statistically significant increase in the possibility of hemoconcentration to be observed (OR = 3.84, 95% CI: [1.24–11.92]) after one week. ✓ The deterioration of renal function was similar in both groups (serum creatinine increased ≥ 0.3 mg/dL).
Packer et al., 2021 a [57]	Multicenter, double-blind, RCT (EMPEROR-Preserved trial)	5988 patients with HF and EF > 40%	Empagliflozin 10 mg/day vs. placebo as an add-on to conventional therapy. Randomization 1:1.	26 months	✓ The outpatient up-titration of furosemide was observed less frequently in the SGLT2 inhibitor arm compared to the control group (HR = 0.76, 95% CI: [0.67–0.86]; *p* < 0.0001). ✓ Treatment with empagliflozin was associated with a 20% to 50% greater likelihood of experiencing a better NYHA functional class after a period of 3 months.
Packer et al., 2021 b [50]	Multicenter, double-blind, RCT (EMPEROR-Reduced trial)	3730 patients with HFrEF (39.6% in “volume overload” and 57% “euvolemic”).	Empagliflozin 10 mg/day vs. placebo as an add-on to conventional therapy. Randomization 1:1.	720 days	✓ Patients with “volume overload” were at a higher risk of increase in the sustained dose of loop diuretics when treated with a placebo vs. empagliflozin (HR: 1.22; 95% CI: [1.00–1.48]; *p* = 0.047). ✓ Empagliflozin administration was associated with a lower probability of the up-titration of loop diuretics (HR = 0.68, 95% CI: [0.55–0.85] in patients with recent volume overload and HR = 0.67, 95% CI: [0.55–0.82] in euvolemic patients. ✓ Patients in the empagliflozin arm experienced a moderate decrease in NT-proBNP levels after 4 weeks and a more significant one at week 52, regardless of volume status (*p* = 0.38 and *p* = 0.67, respectively). ✓ Patients receiving empagliflozin had a higher probability of improvement in the NYHA functional class after 4 weeks and experienced an amelioration in the KCCQ score after 12 weeks of treatment irrespective of their volume status. ✓ A statistically significant decline in body weight (mean 1 kg) and an increase in hematocrit were observed in the group receiving empagliflozin regardless of volume status. ✓ The eGFR decreased at a slower rate in patients receiving empagliflozin regardless of “volume overload”.
Griffin et al., 2020 [59]	Case series (retrospective analysis)	31 patients with ADHF (58% HFrEF) and DR despite loop diuretics administration	SGLT2 inhibitor (canagliflozin or empagliflozin) vs. control as an add-on diuretic therapy	3 days	✓ Body weight significantly decreased in the SGLT2 inhibitor arm compared to the control (mean 1 kg, *p* = 0.03 at day 1; mean 1.7 kg, *p* = 0.08 at day 2; and mean 2.1, *p* = 0.06 at day 3). ✓ SGLT2 inhibitor administration was associated with the elevated excretion of urine (mean 3.7 L, *p* = 0.002 at day 1; mean 3.4 L, *p* = 0.02 at day 2; and mean 3.1 L, *p* = 0.02 at day 3). ✓ The sustained dose of loop diuretics remained stable. ✓ There was no fluctuation in respect to the levels of creatine, blood urea nitrogen, blood pressure, or the occurrence of hypokalemia during the 3 days after SGLT2 inhibitor introduction (*p* = non-significant for all).

### 4.3. Safety Profile and Adverse Effects

The occurrence of common side effects was usually balanced between SGLT-2 inhibitors and placebo comparator treatment arms, and serious adverse effects that led to the discontinuation of the treatment were scarce, suggesting that these drugs are generally well tolerated [42,43,47,50,57,59]. The most common side effects reported were increased urination, genital mycotic infections, urinary tract infections, and volume depletion phenomena [37,43,49,53,54,58,59]. Similar findings have been previously attributed to SGLT-2 inhibitors due to glycosuria [60]. Mild volume depletion (presented with symptomatic hypotension or orthostatic hypotension, polyuria, dehydration, dizziness, vertigo, presyncope, thirst, and rarely orthostatic hypotension), weight loss, a reduction in SBP, the potential deterioration of renal function (increases in serum creatinine levels, decreases in the eGFR), acute kidney injury or failure, potential changes in hematocrit and hemoglobulin, liver function deterioration, diabetic ketoacidosis, and hypoglycemia can also be attributed to the pharmacology of the SGLT-2 inhibitor class [37,50,57,61], which causes osmotic diuresis, natriuresis, glucosuria, and caloric wasting [60]. For instance, in their study, Wilcox et al. reported two cases, one with syncope attributed possibly to orthostatic hypotension and another presented with hypokalemia that required oral potassium chloride to be administered [39]. It is worth noting that in the same study, most of the adverse effects were mild and similar between the patients receiving dapagliflozin and those receiving a placebo [39].

#### 4.3.1. Deterioration of Kidney Function

SGLT-2 inhibitors are considered relatively safe drugs that could actually provide a reno-protective effect rather than harming the kidneys [37,43,49,53,54,57,59,61]. Indeed, although there are a handful of studies that refer to small decreases in the eGFR, which have been observed shortly after the addition of an SGLT-2 inhibitor in the medication of patients hospitalized for ADHF [45,49], the deterioration of renal function was mostly not persistent after hospital discharge [45,49,62]. A trial with a total of 23 patients reported two cases with an acute increase in serum creatinine, which was attenuated later at six weeks [42]. On the other hand, in their study, Schulze et al. demonstrated that markers of renal injury (urinary protein, creatinine, and urinary a1-microglobulin) were similar in both the SGLT-2 inhibitor and the placebo group, suggesting no significant deterioration of the kidneys [43]. In addition, the same study reported no difference between the mean value of the eGFR between patients receiving the SGLT-2 inhibitor and those receiving the placebo [43]. Another study documented a slower rate of eGFR decline in patients receiving empagliflozin [57].

#### 4.3.2. Co-Administration with Other Diuretics

When SGLT-2 inhibitors are given as a standalone treatment, there is a compensatory increase in sodium reabsorption in the distal tubules, which are the target site of thiazides. This occurs after the loop of Henle, and as a result, the actual diuretic effect is constrained. However, the use of SGLT-2 inhibitors in combination with thiazides can lead to substantial diuretic effects [63]. The compensatory reabsorption of sodium in the distal tubules relies on the presence of aldosterone [64]. As a result, when SGLT-2 inhibitors are administered alongside an aldosterone antagonist, it not only inhibits compensatory reabsorption in the distal tubules, but also interferes with reabsorption in the collecting duct (the site where aldosterone antagonists act), resulting in even more enhanced diuresis [65]. Particular attention should be directed towards the findings of the EMPA-REG OUTCOME trial sub-analysis [66], which indicate that the combination of SGLT-2 inhibitors and aldosterone antagonists may worsen HF, potentially due to dehydration and reduced cardiac output [65].

## 5. Gaps in Evidence and Future Research

SGLT-2 inhibitors exhibit substantial efficacy in HF and have demonstrated clear cardioprotective and renoprotective effects. Nonetheless, the exact underlying pathophysiologic mechanisms resulting in their therapeutic benefits remain incompletely understood [67]. Further research is needed in this field [68]. Another aspect requiring further investigation involves determining whether the impact of SGLT-2 inhibitors is influenced by the presence of concurrent cardiovascular comorbidities, as it was recently assessed in a meta-analysis [69]. Moreover, their synergistic effects with other HF medications including diuretics and their combined beneficial outcomes need better comprehension, as well as their application in other diseases and other volume retention conditions apart from acute decompensated HF [70,71]. In addition, it is important to address the long-term outcomes, particularly with the increasing and varied utilization of these medications across different patient populations [72]. Despite the positive effects of SGLT-2 inhibitors in clinical trials, real-world data are also crucial to evaluate their effectiveness and safety in everyday clinical practice, as well as patients’ accessibility to treatment. A recent study demonstrated the underuse of these medications in patients with T2DM and the increased cardiorenal risk [73]. Finally, their impact on other cardiovascular situations that cause myocardial fibrosis and HF, like ST-elevation myocardial infarction or arrhythmias, requires more intensive investigation [74]. Resolving these evidence gaps will lead to more personalized and patient-oriented HF management in the future with improved clinical outcomes.

## 6. Conclusions

The utilization of loop diuretics, a cornerstone in the treatment of acute HF, often encounters diuretic resistance, requiring clinical vigilance and complex management strategies. Clinical evidence suggests that SGLT-2 inhibitors may offer a moderate diuretic effect when added to conventional HF therapy, potentially improving symptoms and fluid balance. Nevertheless, their impact on natriuresis remains uncertain, with studies showing varied responses in urinary sodium excretion. Importantly, the introduction of SGLT-2 inhibitors in HF treatment appears to be generally well tolerated, with manageable adverse effects. Overall, while SGLT-2 inhibitors show promising results in mitigating diuretic resistance in HF, further studies are needed to completely comprehend the underlying mechanisms and optimize their use in clinical practice.

## Figures and Tables

**Figure 1 ijms-25-03122-f001:**
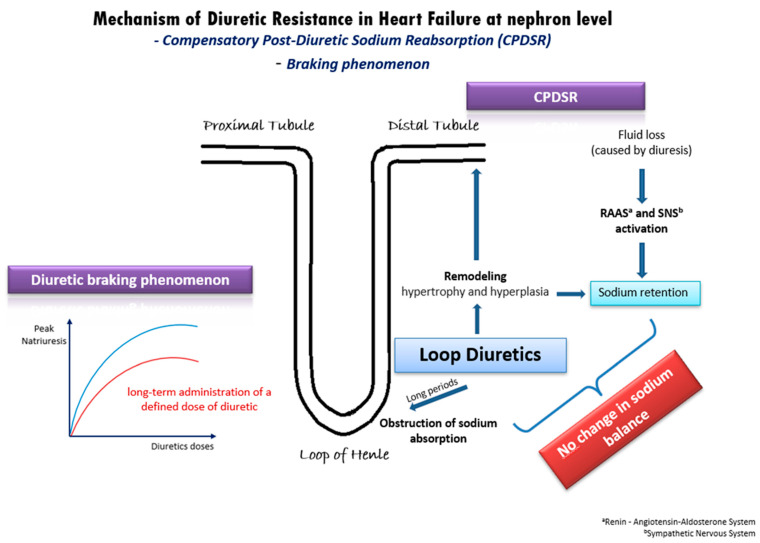
Underlying mechanisms of diuretic resistance in heart failure.

**Figure 2 ijms-25-03122-f002:**
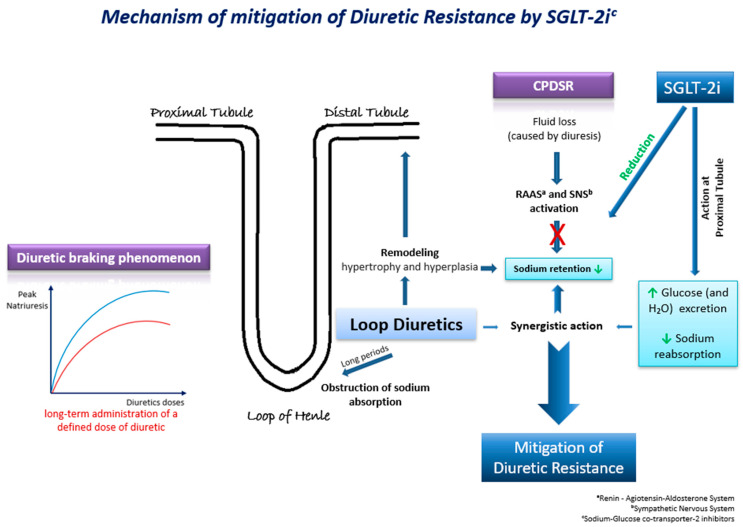
Τhe possible underlying mechanisms through which SGLT-2 inhibitors may mitigate diuretic resistance in HF.

## Data Availability

This is a review article analyzing secondary data, so data sharing is not applicable.

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
