# Peer review of "The Role of Sodium-Glucose Co-Transporter-2 Inhibitors on Diuretic Resistance in Heart Failure"

_ijms, 2024, doi:10.3390/ijms25063122_

Round 1

Reviewer 1 Report

Comments and Suggestions for Authors

This is a well-written narrative review about the possibile role of the SGLT-2 inhibitors on diuretic resistence in heart failure. The discussed topic is of clinical relevance and the gaps in knowledge have been well identified. However, there are some points that authors need to correct:

- In the diuretic resistance paragraph, the braking phenomenon could be explainded in a clearer way

- In the pathophysiology - mechanisms subparagraph, the mechanisms could be explained with more detail

- In the efficacy data, there is a pediod that seems to be little connected to the rest of the text: "At the same time there are several trials reporting an increase in the hematocrit of HF-patients caused by the administration of SGLT-2 inhibitors. The same studies simultaneously suggest that the mechanism of this increase is mostly at tributable to the increase in erythropoietin levels and as a result to erythropoiesis, rather than the hemoconcentration caused by diuresis and volume depletion". And why this period is underlined?

- In the Figure 1 and 2 there are some words underlined in red. 

- Some minor typos in the text need to be corrected.

Comments on the Quality of English Language

The form is correct but minor typos in the text need to be corrected

Author Response

We would like to thank you for the opportunity to resubmit a revised copy of this manuscript. We would also like to take this opportunity to express our thanks to the reviewers for the positive feedback and helpful comments for correction or modification.

We believe we have resulted in an improved revised manuscript, which you will find uploaded alongside this document. The manuscript has been revised to address the reviewer comments, which are appended alongside our responses to this letter. The changes in the revised manuscript have been highlighted in red.

We very much hope the revised manuscript is accepted for publication in International Journal of Molecular Sciences. We are looking forward to hearing from you soon

Reviewer 2 Report

Comments and Suggestions for Authors

Reviewing the review manuscript entitled, “The role of sodium-glucose co-transporter-2 inhibitors on diu-2 retic resistance in heart failure” by Stachteas P et al., This focuses on an effect of SGLT2 inhibitor on diuretic resistance in heart failure. Although this is an interesting review, the point and storytelling are not clear. The pathological conditions and treatment policies for acute heart failure and chronic heart failure are different. The authors should proceed with a narrative discussion after clearly defining the types of heart failure in which DR becomes a problem. For example, the HFrEF described by the authors is a category of chronic heart failure, and the mainstay of treatment is RAS inhibitors and beta-blockers, and natriuresis is not the first-line drug.

 The authors mentioned “despite significant advances in its treatment.” in abstract and the introduction section. This is clearly jumping into conclusions. Although SGLT2 inhibitors and ARNI have been used to treat chronic heart failure, the prognosis of heart failure has not changed significantly.

Comments on the Quality of English Language

There is no major problems.

Author Response

(The authors gave the same response as above.)

Reviewer 3 Report

Comments and Suggestions for Authors

The authors of the current manuscript present a comprehensive review about the effects of sodium glucose co-transporter-2 inhibitors on diuretic resistance in patients with heart failure. The very manuscript is very well structured and thorough. The authors have kept the generally accepted rules for writing such types of articles for medical journals. The title is topical and information provided will be a real contribution to the clinician practice and scientific knowledge in the field. Heart failure is a great medical problem worldwide, affecting between 1 and 2% of the general adult population (and >10% of people, aged >70 years). Although we have elaborated modern strategies - pharmacological and non-pharmacological to treat people with this syndrome, long-term prognosis continues to be unfavorable with almost 50% of all HF patients dying within 5 years after the diagnosis has been established. So, any new information about treatment options, efficacy of different strategies, etc. will be of help to the clinicians.

I recommend the manuscript to be published in its current form.

Author Response

Thank you very much for your encouraging comments about our manuscript.

Round 2

Reviewer 1 Report

Comments and Suggestions for Authors

This is a well-written narrative review about the possibile role of the SGLT-2 inhibitors on diuretic resistence in heart failure. The discussed topic is of clinical relevance and the gaps in knowledge have been well identified. However, i have one more consideration to make:

- In the section "SGLT-2 inhibitors, mechanism of action and current clinical indications" the authors say " In the clinical settings, SGLT-2 inhibitors are employed for the management of T2DM as a second-line therapeutic option subsequent to insufficient glycemic regulation with metformin, or as a primary treatment for individuals at high risk of CVD, HF or CKD.". It should be more emphasized that the use of this class of drugs can be used independently of the glycemic control, citing the recent 2023 ESC guidelines for the management of cardiovascular disease in patients with diabetes (Marx N et al., ESC Scientific Document Group. 2023 ESC Guidelines for the management of cardiovascular disease in patiets with diabetes. Eur Heart J. 2023 Oct 14; doi: 10.1093/eurheartk/ehad192)

Author Response

Thank you very much for the comment. The paragraph have been changed in order to more emphasize the use of this class of drugs independently of the glycemic control according to the recent 2023 ESC guidelines for the management of cardiovascular disease in patients with diabetes (Marx N et al., ESC Scientific Document Group. 2023 ESC Guidelines for the management of cardiovascular disease in patiets with diabetes. Eur Heart J. 2023 Oct 14; doi: 10.1093/eurheartk/ehad192)

Reviewer 2 Report

Comments and Suggestions for Authors

This reaches to an acceptable quality. Congrats.

Author Response

Thank you very much for the useful comments, which substantially help us to reach to an improved version of manuscript.